# *Akkermansia*, a Possible Microbial Marker for Poor Glycemic Control in Qataris Children Consuming Arabic Diet—A Pilot Study on Pediatric T1DM in Qatar

**DOI:** 10.3390/nu13030836

**Published:** 2021-03-04

**Authors:** Arun Prasath Lakshmanan, Amira Kohil, Farah El Assadi, Sara Al Zaidan, Shaikha Al Abduljabbar, Dhinoth Kumar Bangarusamy, Fawziya Al Khalaf, Goran Petrovski, Annalisa Terranegra

**Affiliations:** 1Research Department, Sidra Medicine, Doha P.O. Box 26999, Qatar; szaidan@sidra.org (S.A.Z.); salabduljabbar@sidra.org (S.A.A.); dkbangarusamy@sidra.org (D.K.B.); aterranegra@sidra.org (A.T.); 2Department of Biomedical Sciences, College of Health Sciences, QU Health, Qatar University, Doha P.O. Box 2713, Qatar; ak1404654@student.qu.edu.qa; 3College of Health and Life Sciences, Hamad Bin Khalifa University, Doha P.O. Box 34110, Qatar; felassadi@hbku.edu.qa; 4Endocrinology Clinic, Sidra Medicine, Doha P.O. Box 26999, Qatar; falkhalaf@sidra.org (F.A.K.); gpetrovski@sidra.org (G.P.)

**Keywords:** *Akkermansia*, T1DM, Arabic diet, ethnicity, HbA1c, CSII therapy

## Abstract

In Qatar, Type 1 Diabetes mellitus (T1DM) is one of the most prevalent disorders. This study aimed to explore the gut microbiome’s relation to the continuous subcutaneous insulin infusion (CSII) therapy, dietary habits, and the HbA1c level in the pediatric T1DM subjects in Qatar. We recruited 28 T1DM subjects with an average age of 10.5 ± 3.53 years. The stool sample was used to measure microbial composition by 16s rDNA sequencing method. The results have revealed that the subjects who had undergone CSII therapy had increased microbial diversity and genus *Akkermansia* was significantly enriched in the subjects without CSII therapy. Moreover, genus *Akkermansia* was higher in the subjects with poor glycemic control (HbA1c > 7.5%). When we classified the subjects based on dietary patterns and nationality, *Akkermansia* was significantly enriched in Qataris subjects without the CSII therapy consuming Arabic diet than expatriates living in Qatar and eating a Western/mixed diet. Thus, this pilot study showed that abundance of *Akkermansia* is dependent on the Arabic diet only in poorly controlled Qataris T1DM patients, opening new routes to personalized treatment for T1DM in Qataris pediatric subjects. Further comprehensive studies on the relation between the Arabic diet, ethnicity, and *Akkermansia* are warranted to confirm this preliminary finding.

## 1. Introduction

Type 1 diabetes mellitus (T1DM) is a metabolic disorder, and it is caused by the autoimmune destruction of pancreatic beta cells resulting in insulin deficiency. T1DM affects all age groups irrespective of gender. Based on the International Diabetes Federation (IDF) Diabetes Atlas, the incidence of T1DM continues to increase worldwide, with approximately one million cases presented annually [1], and the diabetic prevalent rate in Qatar is around 17% [2]. T1DM is associated with various other complications, such as severe hypoglycemia, ketoacidosis, diabetic retinopathy, nephropathy, and cardiovascular complications [3]. Despite the severity and the incidence of the disease, the etiopathogenesis of T1DM is still not fully understood, involving a complex interaction between environmental and genetic factors [4]. 

In managing T1DM patients, the therapeutic goal is to manage glucose control, which is accomplished by different treatments, such as insulin therapy and medical nutrition therapy. Insulin therapy (basal-bolus regimen) is one of the recommended approaches in managing patients with T1DM. It is also known as multiple daily injection therapy. Long-acting basal insulin and prandial insulin (rapid-acting) are given for food and corrections as multiple injections at different time intervals throughout the day [5]. The basal insulin dose is based on body weight and insulin sensitivity, whereas the prandial insulin dose is based on carbohydrate intake [6,7]. Basal-bolus therapy has improved glycemic control and reduced the HbA1c level (<7%) in T1DM patients [8]. Insulin pump therapy (also known as Continuous Subcutaneous Insulin Infusion–CSII) involves using an insulin pump device that provides a steady insulin level to the patient. It is considered a common approach used in the out-patient treatment of T1DM [5]. One study showed that episodes of severe hypoglycemia or hyperglycemia were less common in patients under insulin pump treatment [9]. Another study demonstrated that CSII therapy is safe and effective in children and young adults with T1DM. Treatment with CSII therapy has significantly decreased HbA1c levels [10].

There is a strong link between gut microbiota dysbiosis and the development of T1DM, as shown by various animal and human studies. The T1DM patients showed a less-diverse microbiome, with enhanced Bacteroides’ production and reduced level of *Faecalibacterium prausnitzii and Lactobacillus* [11,12]. Various studies have demonstrated that gut microbial dysbiosis and enhanced gut permeability [13] result in beta cell damage and autoimmune activation, contributing to the pathogenesis of T1DM [14,15]. Animal studies have shown that insulin deficiency was associated with the bactericidal dysfunction of Paneth cells, which can alter the intestinal flora [16]. Another approach of T1DM treatment is the mesenchymal stem cell transplantation, which has been reported to deplete the diabetic gut microbiota resistance [17]. However, no clear evidence is available about gut microbiota’s role in affecting the response to the T1DM treatments. For type 2 diabetes mellitus (T2DM), evidence suggests an association between metformin, a widely used anti-diabetic drug, and the gut microbiome, resulting in improved glycemic control. However, the same study showed an insignificant change in microbial diversity between the metformin and the placebo group. Only the acetate, one of the short-chain fatty acids (SCFAs), was significantly different between both groups.

In contrast, butyrate and propinoate showed no statistical difference [18,19]. Furthermore, numerous studies have pointed out that diet, geographical location, and ethnicity play a significant role in shaping individuals’ gut microbiota [20]. However, the association between T1DM treatment and gut microbiota is yet to be demonstrated. No previous study investigated the association between T1DM treatment and gut microbiota in Qatar’s pediatric T1DM population. In this pilot study, we aimed to understand the relationship between the gut microbiota and CSII therapy, evaluating the effect of external factors, such as the diet and nationality, in pediatric T1DM subjects living in Qatar.

## 2. Materials and Methods

### 2.1. Recruitment and Sample Collection

A pilot study recruiting 28 T1DM pediatric patients was conducted at Sidra Medicine hospital in Qatar and approved by the local IRB committee (IRB approval no #1500755). The subjects were approached and introduced to the study after their medical appointment in Sidra’s endocrinology clinic. Study subjects were selected based on the following criteria: being between the age of 6–12 years old, had no medical condition other than T1DM, a disease onset of more than one year, and no history of antibiotic treatment in the previous three months. The subjects and their parents, who agreed to participate in this study, signed parental consent and child assent forms. The physician collected the subject’s clinical information, such as medication, family history of diabetes, insulin treatment, CSII therapy, and diabetes duration during the visit. Also, the dietary intake of the patients was determined by a 24 h food recall during the interview. Anthropometric measurements were also collected, including body weight, height, and the BMI percentile was computed according to age and gender. Stool samples were collected using the OMNIgene®Gut stool collection kit (DNA Genotek, Kanata, ON, Canada ) and stored at −80 degree Celsius until further use.

### 2.2. Bacterial DNA Extraction from Fecal Samples

Bacterial DNA extraction was performed using QIAamp® Fast DNA Stool Mini kit (Qiagen, Germantown, MD, USA) according to the manufacturer’s instructions. The quantity and quality of the extracted DNA were checked by using NanoDrop One (ThermoFisher Scientific, Waltham, MA, USA).

### 2.3. 16SrDNA Library Preparation

The library preparation and the sequencing steps were performed according to the manufacturer’s instructions (MiSeq system, Illumina, San Diego, CA, USA). Briefly, Illumina primers were obtained to target V3-V4 regions of the 16S rDNA gene. The PCR amplifications were carried out under the following conditions: initial at 95 °C for 3 min, 25 cycles at 95 °C for 30 s, 55 °C for 30 s, 72 °C for 30 s, with a final step at 72 °C for 5 min. Then, the secondary amplification (index PCR) was performed using Nextera XT Index Kit (Illumina, San Diego, CA, USA), where dual indices (i7 and i5 indexing primers) and Illumina sequencing adapters (P5 and P7) were added to each sample in the 96-well microplate. The PCR conditions are the same as the one previously mentioned, except the number of amplification cycles is set to 8. The quality of the PCR products was assessed by 2% agarose gel electrophoresis. The library size was detected using Agilent High sensitivity kit (Agilent Technologies, Santa Clara, CA, USA), Agilent 200 Bioanalyzer technology, and quantified using the Qubit dsDNA HS assay kit (ThermoFisher Scientific, Waltham, MA, USA). Finally, the normalized libraries were pooled by mixing 5 μL of the diluted DNA of each library with unique indices in one tube for sequencing. The pooled library and Phix control were denatured using 0.2N NaOH as per the manufacturer’s protocol. Finally, the sample was sequenced using the Miseq reagent v3 kit-600 cycles (Illumina, San Diego, CA, USA) according to the manufacturer’s instructions. Base-calling was directly carried out on the MiSeq.

The raw data were demultiplexed using MiSeq Reporter on Illumina Miseq. PEAR tool was used to merge both forward and reverse end sequences for each sample [21], and the reads with a high-quality score of 30 and above were selected using the Trimmomatic tool [22]. FASTQ files were converted into FASTA files using QIIME v1.9.0 (Quantitative Insights Into Microbial Ecology) pipeline [23]. Operational taxonomic units (OTUs) were obtained by aligning the sequence against the Greengenes database (gg_13_08) with a confidence threshold of 97% [24].

### 2.4. Data Analysis

#### 2.4.1. Clinical Data

The clinical data obtained from the study subjects include gender, nationality, anthropometric measurements (height, weight, BMI percentiles), HbA1c, diabetes duration, and CSII therapy. The two-tails unpaired *t*-test was used to define group differences. *p* < 0.05 was considered statistically significant.

#### 2.4.2. Dietary Data

The dietary data obtained from 24 h food recall was used to evaluate the dietary patterns. The first analysis classified the patients into two groups based on the primary type of consumed foods: Arabic diet, consuming food and recipes from the Arabic culinary tradition; mixed diet consuming mainly or exclusively Western-like food.

#### 2.4.3. Microbiome Data

Alpha diversity and beta diversity analysis were performed using the R package [25,26,27]. Linear discriminant analysis effect size (LEfSe) analysis was used to find out microbial biomarkers among the different groups with the cutoff value of LDA >2.0 [28].

## 3. Results

### 3.1. Study Population and Dietary Habits

We recruited 28 T1DM patients in the age range of 10.5 ± 3.5 years, in which half of the study subjects were Qatari Nationals, and half of the expatriates lived in Qatar. Females accounted for 35.7% of the study subjects, and males accounted for 64.2%, with a mean BMI percentile of 57.59 ± 29.92 in the range of the normal weight. The HbA1c average level and the diabetes duration for the total study subjects were 9.75 ± 1.62% and 8 ± 4.24 years, respectively. Out of the 28 participants, only 11 subjects had undergone CSII therapy. Based on the dietary pattern, 56% of the study subjects consumed an Arabic diet, whereas 42.8% consumed a mixed Western-like diet. Most of the Qatari Nationals (62.5%) consumed the Arabic diet, compared to the expatriates consuming mostly a mixed Western-like diet (66%). The baseline characteristics of the study subjects are shown in Table 1. 

### 3.2. Effect of CSII Therapy and HbA1c Level on the Gut Microbiome Composition in the T1DM Subjects

We have measured the relative abundance of gut microbiota in our T1DM study subjects. Alpha diversity analysis has revealed that the microbial abundance level (measured by Simpson) was higher in the CSII group than the non-CSII group. Still, there was no statistically significant difference in the genus richness (measured by Observed and Chao1) between these two groups (Figure 1a). Moreover, beta diversity analysis has shown no statistical difference between these two groups (Figure 1b). The significant bacteria at the phyla level were Bacteroidetes, Firmicutes, Actinobacteria, Proteobacteria, and Verrucomicrobia (Appendix A). We then evaluated CSII therapy’s effect and the HbA1c level on the gut microbial composition in the T1DM subjects. However, the identification of gut microbial markers using the LEfSe analysis has demonstrated that the phylum Verrucomicrobia and particularly the genus *Akkermansia* were significantly enriched in the T1DM patients without CSII therapy.

In contrast, the family Christensenellaceae, order *Neisseriales,* and genus *Klebsiella, Escherichia, Pseudobutyrivibrio,* and *Aggregatibacter* were increased in subjects who had undergone CSII therapy (Figure 2a). Interestingly, *Akkermansia* was found to be enriched in the subjects with poor glycemic control (>7.5%) than the subjects with better glycemic control (<7.5%) (Figure 2b).

### 3.3. Impact of Diet on the Abundance of the Genus Akkermansia in the Poorly Controlled T1DM Subjects 

We investigated the impact of diet on the abundance of the phylum Verrucomicrobia and the genus *Akkermansia* in the T1DM subjects with poor glycemic control and the CSII therapy. Phylum Verrucomicrobia and the genus *Akkermansia* significantly enriched in the Arabic-diet-consuming subjects without the CSII therapy (Figure 3a). In addition to this, *Akkermansia* level was higher only in the poorly controlled subjects having a high level of HbA1c (>7.5%) and consuming the Arabic diet (Figure 3b). Interestingly, subjects with a high level of HbA1c had an elevated level of the genus *Ruminococcus* (Figure 2b), and the subjects consuming the Arabic diet and with a high level of HbA1c also showed an elevated level of *Ruminococcus* (Figure 3b).

### 3.4. Impact of Nationality on the Abundance of the Genus Akkermansia in the Poorly Controlled T1DM Subjects

Interestingly, we have found that the phylum Verrucomicrobia and genus *Akkermansia* were enriched only in the Qataris subjects without the CSII therapy, but not in the expatriates (Figure 4a). When we classified the subjects with the combination of HbA1c and nationality, surprisingly, we did not see an enrichment of *Akkermansia* in any group (Figure 4b).

### 3.5. Influence of Arabic Diet in Qataris Subjects on the Abundance of the Genus Akkermansia in the T1DM Subjects

Here, we have intended to explain the role of Qataris nationality and the Arabic diet on the abundance of *Akkermansia* in the study population. Interestingly, we have found that Qataris pediatric T1DM subjects had elevated levels of *Akkermansia* compared to their counterpart (Figure 5a). Surprisingly, we did not find an enriched *Akkermansia* level when we classified the subjects based only on the diet pattern (Figure 5b).

Finally, when we analyzed the data based on both nationality and the dietary pattern, only Qataris pediatric T1DM subjects who consumed the Arabic diet had shown a higher abundance of *Akkermansia* among the other groups (Figure 6).

## 4. Discussion

In this pilot study, we aimed to identify the effect of CSII therapy and the HbA1c level on the gut microbiome composition and the impact of diet and nationality in the T1DM pediatric subjects living in Qatar. To the best of our knowledge, this is the first study that showed the genus *Akkermansia* as a potential microbial marker of insulin treatment response. Its abundance is affected by the nationality and the diet in the pediatric T1DM subjects living in Qatar. It has been well established that both the T1DM and gut microbiome regulate each other in humans and animal models [29,30,31,32]. The relationship between these two factors is complex in nature. Studies have indicated that one of the earliest indications for dysbiosis in T1DM is the Bacteroidetes/Firmicutes ratio changes and the reduced diversity index in the pediatric T1DM subjects [33,34]. Interestingly, in this study, we did not find any significant differences in the Bacteroidetes/Firmicutes ratio.

The classical treatment strategy for the maintenance of normoglycemia in the T1DM subjects is the administration of insulin. Insulin can be administered via single-dose injection or multiple-dose injections, depending on the insulin dose requirement. Insulin dose is determined based on the body weight and the carbohydrate coverage, i.e., insulin-to-carbohydrate ratio [35,36]. In addition to this, CSII therapy in children is getting good acceptance by the healthcare providers. The SWEET registry cohort data has shown that the children who underwent CSII therapy had achieved reasonable metabolic control compared to the children without CSII therapy [37]. In this study, initially, we aimed to find the effect of CSII on the gut microbiome of pediatric T1DM subjects, and the results have shown that the genus *Akkermansia* was significantly enriched in the subjects without the CSII therapy. Moreover, genus abundance measured by the Simpson method has shown that the subjects with CSII therapy had higher abundance than their counterpart. We aimed to measure the microbiome composition in poorly controlled patients (HbA1c level > 7.5%). HbA1c test estimates a patient’s three-month average blood glucose level and is a widely accepted measure in diabetic subjects to evaluate their glycemic control [38]. Surprisingly, *Akkermansia* was significantly enriched in the poorly controlled group compared to the controlled group. Indeed, it is fascinating because *Akkermansia muciniphila (A. muciniphila)*, a mucin degrading bacterium, has been widely reported to play a significant role in promoting the gut barrier function, epithelial cell integrity, and enhancement of trans-epithelial resistance through various mechanistic pathways such as the expression of genes related to the immune response, normalizing the metabolic endotoxemia and lipid metabolism [39,40,41,42,43,44]. *A. muciniphila* has been found to mediate the metformin’s beneficial effects, such as the regulation of glucose metabolism via the SCFAs production in human and mice studies [45,46]. Few contrary reports suggest that the *A. muciniphila* level was elevated during the progression of chronic kidney diseases (CKD) [47]. The Bio-Breeding Diabetic Prone (BBdp) rats have shown that mucin reduction increases the susceptibility to T1DM [48]. In addition to this, administration of *A. muciniphila* grown-up in a mucin-less environment has a significant reduction effect on the obesity in HFD-induced obesity in mice than the *A. muciniphila* grown-up in a mucin-rich environment. So, it is evident that the mucin content plays a pivotal role in the gut barrier function [49]. However, *A. muciniphila* in most diseases seem to play protective roles and in a few conditions like T1DM and CKD, where it might play a controversial role. More comprehensive studies are needed to clarify the exact role of *A. muciniphila* in the T1DM subjects.

Diet and environment can play a role in explaining the controversial findings around *A. muciniphila.* Diet is known to modulate *Akkermansia* level, as demonstrated in animal models where the Western high-fat diet drastically reduced the abundance of *Akkermansia* in Apo^e-/-^ mice, restored only after the supplementation of *A. muciniphila* [50]. In agreement with this study, we have found that the *Akkermansia* was significantly enriched in the subjects without CSII therapy and non-controlled HbA1c level (>7.5%) consuming the Arabic diet compared to the mixed Western-like diet. Moreover, previous evidence suggests that the abundance of *A. muciniphila* is positively associated with the family *Ruminiococcaceae,* genus *Gordonibacter,* and species *Methanobrevibacter smithii* [51,52]. In this study, we also found that the genus *Ruminococcus* was elevated along with the *Akkermansia* in the subjects who had high HbA1c levels and consumed the Arabic diet.

Furthermore, we evaluated the impact of nationality on the abundance of *Akkermansia*. Interestingly, our data revealed that only the Qatari pediatric T1DM subjects without the CSII therapy conserved the microbial profile characterized by the abundance of *Akkermansia* compared to the expatriate’s T1DM subjects living in Qatar. It suggests that host genetic factors might be involved, as previously reported, where the host genetic background could modulate the gut microbiota composition [53,54]. A recent study on monozygotic (MZ) and dizygotic (DZ) twins has shown that MZ twins had greater microbial similarities than the DZ. The different gut microbial compositions were observed in mice with mutations in genes involved in the inflammatory pathways and diabetes compared to their wild-type counterparts [55,56]. To test this hypothesis, we have analyzed the gut microbial composition based on nationality. Strikingly, the results showed that the *Akkermansia* is significantly enriched only in the Qatari pediatric T1DM subjects, not in the expatriate’s T1DM subjects. So, we cannot rule out the possibility of host genetic factors for an elevated level of *Akkermansia* in the pediatric T1DM subjects. When we analyzed the gut microbial composition data based solely on the dietary pattern, it has revealed no apparent difference of *Akkermansia* between the subjects who consumed the Arabic diet compared to the mixed Western-like diet. So, there could be a possibility that host genetic factors might have a more pronounced effect on the gut microbial modulation than the dietary factors.

Interestingly, when we classified the gut microbial abundance based on the combination of two factors, i.e., a diet with nationality, we have seen the enrichment of *Akkermansia* only in the Qataris T1DM subjects who consumed the Arabic diet. So, the dietary pattern is still playing its part in modulating the *Akkermansia* level in the Qataris pediatric T1DM subjects. Appendix A shows the relative abundance level of the genus *Akkermansia* in the subjects classified at various combinations and, moreover, when we compared the relative abundance of *Akkermansia* with the healthy control (HC) subjects, we have found that there was a significant increase (*p* = 0.0097) in the abundance level of genus *Akkermansia* in T1DM subjects Appendix A (j), and there could be a possibility that the occurrence of T1DM would increase the rate of mucous secretion in these pediatric subjects. 

The Qataris population has a unique genetic background, as confirmed from the Qatar Genome Project (QGP) that identified 58.37% novel genetic variants in Qataris. It should be considered in interpreting the data [57]. In support of this concept, a pilot study from an Italian group suggested that the gut microbiota profiles differ among individuals depending on their region of origin [58]. In support of this concept, A large cohort study conducted by Turpin et al. (2016) has shown that the gut microbiome could have greater association with the host genetic factor [59]. Composition of gut microbiota is mainly influenced by both environmental and genetic factors [60] in T1DM. In recent times, researchers with the help of sophisticated computational analysis established that one of the main focuses in the treatment of T1DM is understanding the origin of gut microbial dysbiosis. The contribution of the host genetic factor in shaping the gut microbial composition was well demonstrated by Mullaney et al. (2018) and others using non-obese diabetic (NOD) mice animal model [61,62]. They showed that NOD mice carry the protective alleles of T1DM susceptibility loci, namely, major histocompatibility complex (MHC) and *Idd3* and *Idd5*, in which *Idd3* and *Idd5* improves the regulatory T cells (Tregs) function and immune tolerance along with the distinct gut microbial composition with an increased abundance of *Akkermansia*. In addition to this, environmental factors, such as diet, infection, and usage of antibiotics, can have more a profound effect on the pathogenesis of T1DM. For example, Zhang et al. (2018) demonstrated that single-course usage of tyrosine tartrate (macrolide antibiotic) accelerates the occurrence of T1DM in male NOD pups, suggesting a crucial link of antibiotics in early life [63]. On the contrary, recent clinical and pre-clinical studies suggested that not all antibiotics impact on pathogenesis of T1DM, especially in shaping the gut microbiome in early life period [64,65]. Furthermore, usage of vancomycin increased the abundance of *Akkermansia* and modulates the glucose metabolism in mice [66]. Finally, shaping of gut microbiota in children, particularly at a younger age might be affected with various environmental factors and genetic factors, and careful interpretation is needed for the treatment of T1DM.

## 5. Conclusions

In this pilot study, we found that the genus *Akkermansia*, a mucin-degrading bacterium, was significantly elevated in the Qatari pediatric T1DM subjects with poorly controlled HbA1c levels (>7.5%) and consuming an Arabic diet. A combined effect of host genetic factors along with the dietary habits can explain the elevated level of *Akkermansia* leading to higher mucous production. Novel genetic variants found in the Qatari population [57] could contribute to an increase in the level of *Akkermansia* in pediatric T1DM subjects. In addition, the consumption of an Arabic diet at a younger age might play a role in modulating the level of *Akkermansia*, since dietary pattern at a younger age is involved in shaping the gut microbiome. So, there is a possibility that the combination of genetic and dietary factors could play significant roles in defining or shaping the gut microbiota, especially in pediatric T1DM subjects. All these hypotheses warrant a larger cohort study using Qatari pediatric subjects to better understand the interaction of host genetic factors and diet on the level of *Akkermansia* in pediatric T1DM subjects.

### Study Limitation

Our study comes with some limitations, mainly due to the small number of subjects and the use of 24 h diet recalls for diet analysis that reporting a one-day diet may not represent the patients’ habits. Another limit is that it is impossible to further classify the expatriates according to the country of their origin. Our study’s strength is the well-selected group of patients, with a confirmed diagnosis of T1DM and a good number of subjects with CSII therapy compared with subjects using insulin injection. We tried to mitigate the uncertainties of dietary records recruiting patients in a narrow age range (6–12 years old) that exclude infants, still under breastfeeding and complimentary food and with an immature microbiome, and teenagers with a wide range of social and dietary habits. Larger studies are needed to confirm the preliminary findings of this pilot study, inclusive of detailed dietary intakes and nationality information, ideally correlated by genetic analysis.

## Figures and Tables

**Figure 1 nutrients-13-00836-f001:**
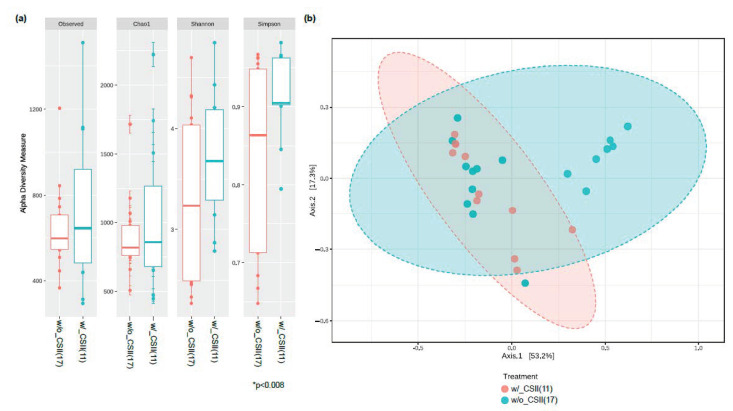
Effect of CSII therapy on the gut microbiome composition in the T1DM subjects. (**a**) The gut microbial diversity index measured by alpha diversity revealed that the subjects who had CSII therapy had higher microbial abundance (only in Simpson index). (**b**) There were no microbial dissimilarities (beta-diversity) in subjects with or without the CSII therapy. β-Diversity was visualized using Principle co-ordinations generated with the Bray–Curtis distance metric using QIIME. Analysis of group similarity (ANOSIM) was measured between categories included in this study using 1000 permutations, * *p* < 0.05. (**b**) Note: g UC, genus_unclassified; w/_CSII, with CSII therapy; w/o_CSII, without CSII therapy.

**Figure 2 nutrients-13-00836-f002:**
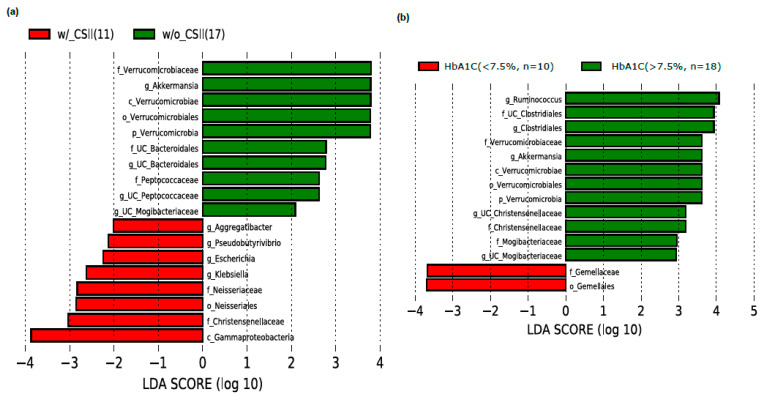
Effect of CSII therapy and HbA1C level on the gut microbiome composition in the T1DM subjects. The LEfSe analysis showed (**a**) significant enrichment of *Akkermansia* in the subjects without CSII therapy and (**b**) significant enrichment of *Akkermansia* in the subjects with HbA1c level of more than >7.5% and <7.5% LDA cutoff value >2.0. Note: g_UC, genus_unclassified.

**Figure 3 nutrients-13-00836-f003:**
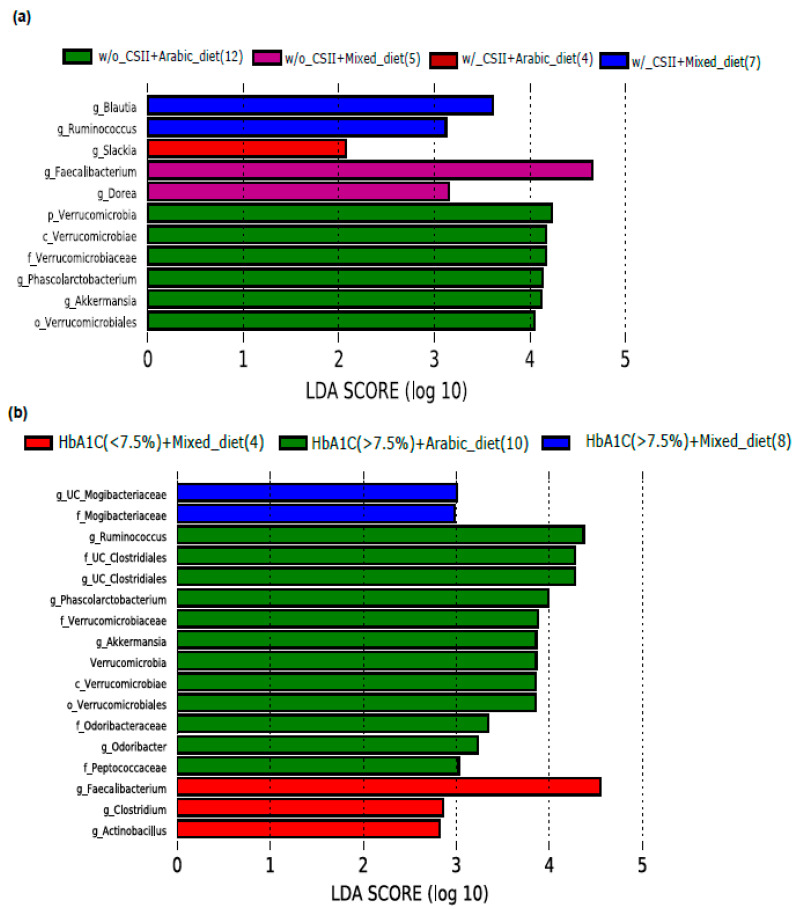
Impact of CSII therapy and diet on genus *Akkermansia* in poorly controlled T1DM subjects. (**a**,**b**) LEfSe analysis revealed that *Akkermansia* was enriched only in the subjects without CSII therapy treatment and poor glycemic control who consumed the Arabic diet. LDA cutoff value >2.0; Note: g_UC, genus_unclassified; w/_CSII, with CSII therapy; w/o_CSII, without CSII therapy.

**Figure 4 nutrients-13-00836-f004:**
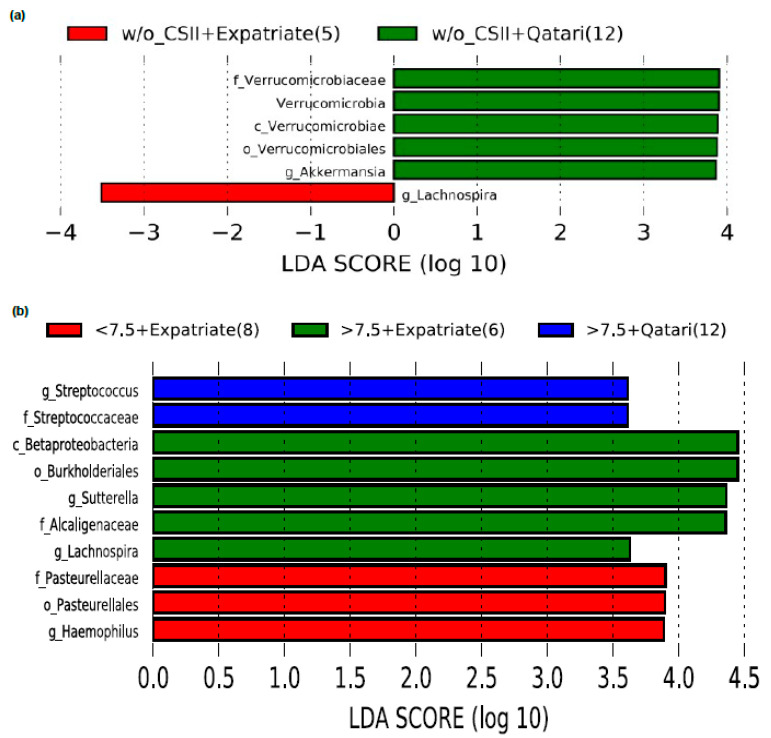
Impact of CSII therapy and nationality on genus *Akkermansia* in poorly controlled T1DM subjects. (**a**) *Akkermansia* was enriched only in the Qataris T1DM subjects without CSII therapy and (**b**) *Akkermansia* disappeared analyzing HbA1c levels according to nationality. LDA cutoff value > 2.0; Note: g_UC, genus_unclassified; w/_CSII, with CSII therapy; w/o_CSII, without CSII therapy.

**Figure 5 nutrients-13-00836-f005:**
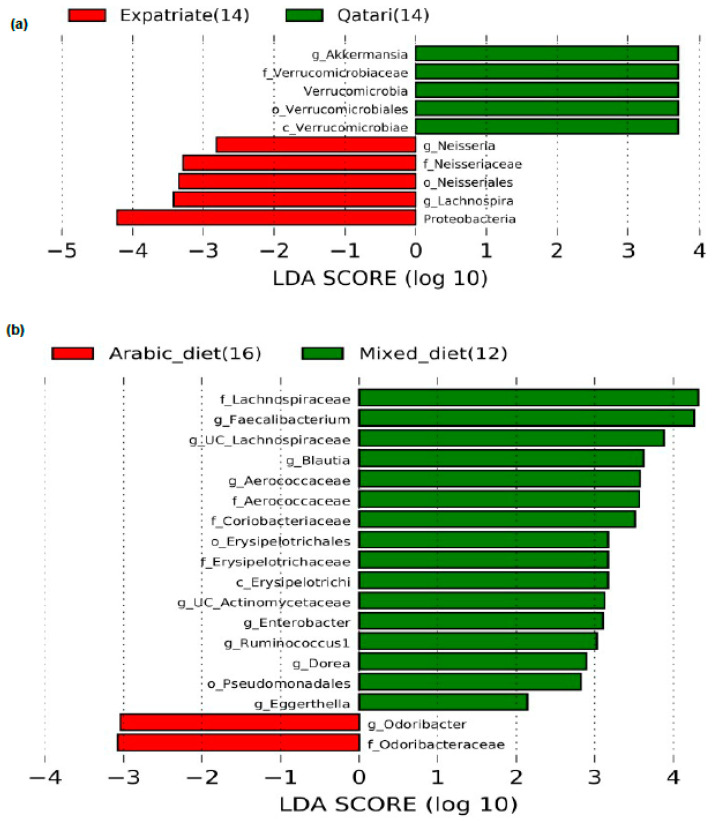
Impact of diet and nationality on genus *Akkermansia* in T1DM subjects. LEfSe analysis has revealed that (**a**) the *Akkermansia* was only enriched in the Qataris T1DM subjects, and (**b**) *Akkermansia* was not significantly different between the Arabic diet and mixed diet consumed subjects. LDA cutoff value > 2.0 g_UC represents genus_unclassified.

**Figure 6 nutrients-13-00836-f006:**
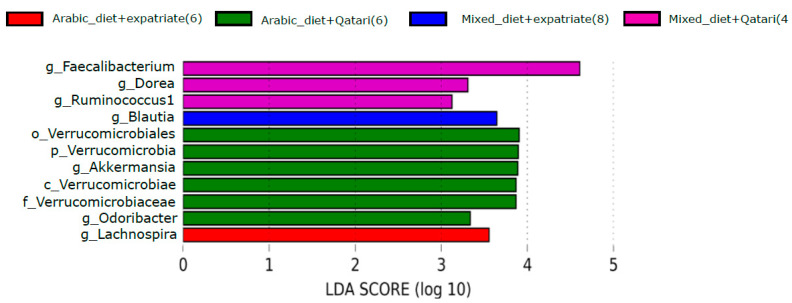
Impact of combination of diet and nationality on genus Akkermansia in T1DM subjects. LEfSe analysis has revealed that the *Akkermansia* was only enriched in the Qataris T1DM subjects, particularly in Qataris who consumed the Arabic diet. LDA cutoff value > 2.0 g_UC represents genus_unclassified.

**Table 1 nutrients-13-00836-t001:** General characteristics of the study participants.

Parameters	T1DM Patients
Number of subjects	28
Age in years	10.50 ± 3.53
Gender	
Female	10 (39.3%)
Male	18 (60.7%)
Dietary pattern	
Arabic diet	16 (57%)
Mixed Western-like diet	12 (42.8%)
Nationality	
Qatari	14 (50%) (Arabic diet = 10, mixed diet = 4)
Expatriate	14 (50%) (Arabic diet = 6, mixed diet = 8)
HbA1c (%)	9.75 ± 1.62
<7.5%	10 (35.7%) (Arabic diet = 06, mixed diet = 04)
>7.5%	18 (64.3%) (Arabic diet = 10, mixed diet = 08)
CSII	
Yes	11 (39.3%) (Arabic diet = 04, mixed diet = 07)
No	17 (60.7%) (Arabic diet = 12, mixed diet = 05)
Diabetes Duration (years)	8.00 ± 4.24
BMI Percentile	57.59 ± 29.92

Note—The values in the table are represented as mean ± SD and in percentage, wherever applicable. CSII: Continuous Subcutaneous Insulin Infusion; HbA1c: Hemoglobin A1c.

## Data Availability

The data presented in this study can be found here in the NCBI’s Bio project repository: [PRJNA693107 and PRJNA701159].

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
