# Peer review of "Akkermansia, a Possible Microbial Marker for Poor Glycemic Control in Qataris Children Consuming Arabic Diet—A Pilot Study on Pediatric T1DM in Qatar"

_nutrients, 2021, doi:10.3390/nu13030836_

Round 1
Reviewer 1 Report
This study investigates the relationship between the gut microbiome and CSII therapy, dietary habits, as well as HbA1c levels in pediatric T1DM subjects from Qatar. Stool samples were collected from 28 patients with T1DM and the relationship between patient background and their microbiome was analyzed by discriminant analysis. The authors found Akkermansia muciniphila was enriched in T1DM patients from Qatar with an Arabic diet.
The hypothesis that host genetic factors and diet might play a significant role in the elevation of A. muciniphila levels in T1DM patients is intriguing. Nonetheless, I think the manuscript could be further improved by making some minor revisions.
- Because all subjects were T1DM patients, it is necessary to consider the difference of microbiome between T1DM patients and healthy individuals in Qatar.
- The authors refer to enrichment of Akkermansia muciniphila, but all the data are given at the genus level. Thus, “ A. muciniphila” should be changed to “genus Akkermansia” throughout the manuscript. If the authors wish to show the enrichment of Akkermansia muciniphila, the results for Akkermansia muciniphila from the obtained sequence data in this study should be given. Alternatively, if possible, the data should be confirmed using another method such as qPCR.
- The authors indicated the genus Akkermansia is the most important microbe in this study. I think it would be better to show the relative abundance of the genus Akkermansia in each group as an independent graph.
- The authors may need to further consider the confounding factors. Twelve out of 14 Qataris patients were w/o CSII and 12 were HbA1c >7.5. The results given in Fig. 1 and Fig. 2 seem to be influenced by nationality (genetic) bias.
- The font size used in the figures is too small to read.
- In Fig. 1, the figure legend does not match the figure numbers.
- In Fig. 1(c), the authors should describe the method of dimensional contraction and distance.
- In Fig. 2(a), group names should be corrected. Specifically, the group name given in blue and purple is the same.
- In lines 164, 183, “Verrucomicrobia phylum” should be corrected to “phylum Verrucomicrobia”.
- All microbe names should be added to the taxonomic rank in front of each corresponding name.
- The ethical information in this clinical study should be described in the “Institutional Review Board Statement”.
Author Response
Please see the attachment also.
Response to Reviewers
Reviewer #1
The hypothesis that host genetic factors and diet might play a significant role in the elevation of A. muciniphila levels in T1DM patients is intriguing. Nonetheless, I think the manuscript could be further improved by making some minor revisions.
Q1: Because all subjects were T1DM patients, it is necessary to consider the difference of microbiome between T1DM patients and healthy individuals in Qatar.
Response: We agree with your point. Since, our main aim was to understand the relationship between the gut microbiota and CSII therapy, evaluating the effect of external factors, such as the diet and nationality, in pediatric T1DM subjects, and we thought comparing microbiome from the T1DM subjects with healthy control subjects would not yield any interesting facts due to the fact that the healthy controls are non-treated subjects. However, we have measured the relative abundance of the most important bacteria in this study, i.e. genus Akkermansia, and compared with the T1DM subjects. Intriguingly, we did find that there was a statistically significant (**p=0.0097) difference in the abundance of genus Akkermansia in the T1DM subjects than the healthy control subjects. The data was included in the supplementary file Figure 2S (j).
Q2: The authors refer to enrichment of Akkermansia muciniphila, but all the data are given at the genus level. Thus, “ A. muciniphila” should be changed to “genus Akkermansia” throughout the manuscript. If the authors wish to show the enrichment of Akkermansia muciniphila, the results for Akkermansia muciniphila from the obtained sequence data in this study should be given. Alternatively, if possible, the data should be confirmed using another method such as qPCR.
Response: As per your suggestion, we have changed A. muciniphila to Akkermansia throughout the manuscript and it is highlighted in the revised manuscript.
Q3: The authors indicated the genus Akkermansia is the most important microbe in this study. I think it would be better to show the relative abundance of the genus Akkermansia in each group as an independent graph.
Response: Thank you for your valuable suggestion. As per your comments, we have shown the relative abundance of the genus Akkermansia in each group as an independent graph and it was shown in the supplementary figure 2S.
Q4: The authors may need to further consider the confounding factors. Twelve out of 14 Qataris patients were w/o CSII and 12 were HbA1c >7.5. The results given in Fig. 1 and Fig. 2 seem to be influenced by nationality (genetic) bias.
Response: Thank you for your valuable comment and we completely agree with your point. We could not completely disregard the fact that the data presented (Fig.1 and Fig. 2) in this pilot study might be influenced by the nationality (genetic) bias, and this could be due to small number of subjects that we recruited at this point of time. However, we cannot rule out the possibility, that is genus Akkermansia might be unique to the Qatari subjects, because when we analyze the relative abundance of Akkermansia between w/o_CSII (expatriate, n=5) and w/o_CSII (Qatari, n=12), and between HbA1c>7.5% (expatriate, n=6) and HbA1c>7.5% (Qatari, n=12) [supplementary figure. 2S (e) and (f)], it seemed to be in the higher proportion in Qatari subjects. This paradox is of great interest to us and we are planning to conduct a big cohort to confirm this in our future study.
Q5: The font size used in the figures is too small to read.
Response: As per your comment, in the revised manuscript, we have improved the font size in each figure by increasing the size of the figure and limiting the number of data in each figure. The changes are highlighted in the revised manuscript.
Q6: In Fig. 1(c), the authors should describe the method of dimensional contraction and distance.
Response: As per your valuable comment, we have described the method of dimensional contraction and distance and mentioned in the figure legends (Fig. 1b) in the revised manuscript.
Q7: In Fig. 2(a), group names should be corrected. Specifically, the group name given in blue and purple is the same.
Response: As per your comment, we have corrected the mistake and changes are highlighted in the revised manuscript.
Q8: In lines 164, 183, “Verrucomicrobia phylum” should be corrected to “phylum Verrucomicrobia”.
Response: As per your comment, we have corrected the mistake in the revised manuscript.
Q9: All microbe names should be added to the taxonomic rank in front of each corresponding name.
Response: As per your comment, we have added the taxonomic rank in the revised figure.
Q10: The ethical information in this clinical study should be described in the “Institutional Review Board Statement”.
Response: As per your valuable suggestion, we have added ethical approval number in the methodology section.

Reviewer 2 Report
In the article entitle Akkermansia muciniphila, a Possible Microbial Marker for Poor 2 Glycemic Control in Qataris Children Consuming Arabic Diet 3 – A Pilot Study on Pediatric T1DM in Qatar Authors have taken under consideration the important problem T1DM.
The subject of their studies were young persons around 10 years old. In my opinion, the clinical group has been chosen correctly, however was too small for significant statistical meaning. From the medical point of view, the gut microbiome discussion in the diabetes type one or two contexts is important. Moreover, taken into account the diet factor influence on the microbiome in the case of adolescents can give the response in the quality of life of adults. Therefore, the results described in this article are valuable for a broad spectrum of readers. It would be a wonderful if the Authors would like to extend the conclusion part which is now not acceptable.
Author Response
Please see the attachment also
Response to Reviewer #2
Reviewer#2
In the article entitle Akkermansia muciniphila, a Possible Microbial Marker for Poor 2 Glycemic Control in Qataris Children Consuming Arabic Diet 3 – A Pilot Study on Pediatric T1DM in Qatar Authors have taken under consideration the important problem T1DM.
Q1: The subject of their studies were young persons around 10 years old. In my opinion, the clinical group has been chosen correctly, however was too small for significant statistical meaning. From the medical point of view, the gut microbiome discussion in the diabetes type one or two contexts is important. Moreover, taken into account the diet factor influence on the microbiome in the case of adolescents can give the response in the quality of life of adults. Therefore, the results described in this article are valuable for a broad spectrum of readers. It would be a wonderful if the Authors would like to extend the conclusion part which is now not acceptable.
Response: Thank you for your valuable suggestion. We completely agree with your suggestion. As per your comment, we have revised our discussion and conclusion part in the revised manuscript.
